# Research on Multi-Agent D2D Communication Resource Allocation Algorithm Based on A2C

**Xinzhou Li, Guifen Chen \*, Guowei Wu, Zhiyao Sun and Guangjiao Chen**

School of Electronic and Information Engineering, Changchun University of Science and Technology, Changchun 130022, China

\* Correspondence: 2019100482@mails.cust.edu.cn

**Abstract:** Device to device (D2D) communication technology is the main component of future communication, which greatly improves the utilization of spectrum resources. However, in the D2D subscriber multiplex communication network, the interference between communication links is serious and the system performance is degraded. Traditional resource allocation schemes need a lot of channel information when dealing with interference problems in the system, and have the problems of weak dynamic resource allocation capability and low system throughput. Aiming at this challenge, this paper proposes a multi-agent D2D communication resource allocation algorithm based on Advantage Actor Critic (A2C). First, a multi-D2D cellular communication system model based on A2C Critic is established, then the parameters of the actor network and the critic network in the system are updated, and finally the resource allocation scheme of D2D users is dynamically and adaptively output. The simulation results show that compared with DQN (deep Q-network) and MAAC (multi-agent actor–critic), the average throughput of the system is improved by 26% and 12.5%, respectively.

**Keywords:** D2D communication; deep reinforcement learning (DRL); interference management; spectrum resource allocation; power control

## 1. Introduction

With the rapid development of the fifth-generation mobile communication technology and the arrival of the new era of We Media, the number of intelligent terminal devices has increased rapidly [1,2]. The Internet of Things household equipment [3], autonomous vehicle, [4], smart roads, smart cities [5], etc. have entered thousands of households. The accompanying mobile network online conference, 4K ultra clear audio and video programs, mobile We Media, Metauniverse, cloud storage, cloud services and other applications occupy more and more cellular network services [6]. High speed, large capacity, low delay and other communication requirements are constantly put forward, data traffic is experiencing explosive growth, and cellular network technology is advancing. According to statistics, globally, the number of intelligent terminal devices and the number of interconnections (CAGR of 10%) are growing much faster than the population (CAGR of 1.0%) and Internet users (CAGR of 6%). This trend has promoted the growth rate of the number of intelligent terminals owned by each family and each person [7]. Cisco forecasts that the number of network devices per capita will reach 3.6. With a variety of new terminals with different forms, functions and uses entering the market, more and more applications, such as medical care, intelligent security, transportation logistics and smart homes have exploded [8,9]. They not only improve labor productivity, but also improve people's life experience, but also increase the traffic burden on wireless communication networks [10].

According to the statistics of the Internet Data Center (IDC), in the next 10 years (2015–2025), the CAGR of new data generated each year will be about 26%. It is predicted that the new data traffic will reach 175.8 zettabytes (ZB) in 2025, an increase of nearly ten

times compared with 18.2 ZB in 2015 [11]. This has led to the severe congestion of traditional cellular base stations designed for large-radius service areas, especially in campus, government and other intensive office spaces and large business districts [12]. During commercial or holiday celebrations, the rapidly increasing amount of communication equipment will impose a huge burden on the limited spectrum resources. At the same time, the development of communication technology will be difficult by 2030. It consumes 51% of the world's electricity and exacerbates global greenhouse gas emissions, accounting for 23% of the total [13]. If effective control measures are not taken, energy and environmental problems will affect the global economy and threaten human physical and mental health.

Therefore, on the one hand, the future cellular network needs to improve the utilization of spectrum resources and ensure the reliable communication of user communication. On the other hand, it needs green and reliable communication technology to mitigate energy consumption. As the main component of the next generation of communication, D2D communication technology has attracted much attention [14].

D2D communication technology refers to enabling two terminal devices with close geographical locations to bypass the base station (BS) directly for short-distance communication [15]. In a broad sense, any link that can communicate directly in authorized or unauthorized frequency bands can be regarded as D2D communication [16]. Due to the inherent requirements for high-speed data transmission, low-delay communication and specific areas' or users' communication quality, D2D communication technology is considered the key technology in 5G cellular networks and an important part of future communication [17]. D2D communication can also save energy. In short-distance communications, the transmission power of D2D is very small, which greatly saves electric energy. In terms of the use of spectrum resources, D2D communication has a variety of communication modes. In the D2D communication mode, only one communication link is required for information transmission; two communication links are required in cellular mode. Most importantly, cellular users and D2D users can share the same radio spectrum resource.

By introducing D2D communication technology into wireless cellular networks, nearby intelligent terminals can directly establish direct communication links. On the one hand, this can improve the spectrum utilization in wireless communication networks; on the other hand, it can greatly improve the data throughput and user experience of dense users. However, when multiple D2D users reuse cellular users, although they gain many advantages, they also bring about link interference that cannot be ignored. This interference includes same-layer interference between D2D users when multiple D2D communication users reuse the same cellular user, and cross-layer interference between D2D users and cellular users. The same-layer interference and inter-layer interference greatly limit the development of D2D communication technology.

In order to reduce the interference in D2D communication systems, some traditional resource allocation schemes are proposed. These methods can be divided into centralized and distributed.

Centralized resource allocation mainly includes resource allocation based on graph theory [18], resource allocation based on airspace isolation [19] and some resource allocation schemes based on a meta heuristic algorithm. Distributed resource allocation mainly includes resource allocation based on game theory [20,21] and resource allocation based on machine learning [22]. In reference [23], a radio resource allocation scheme based on imperfect channel information of artificial bees was proposed. The artificial bee colony algorithm was used to optimize the distance ratio between cellular users and D2D users, select the optimal resource matching scheme and improve the connection number of D2D users and the system throughput. Reference [24] proposed a tabu search (TS)-based D2D communication resource allocation meta heuristic algorithm to solve power allocation and RB allocation problems with lower complexity. Compared with similar algorithms, the algorithm complexity is lower than that of genetic algorithms and maximizes the system's security capacity under the minimum quality of service requirements. Reference [25] pro-

posed the maximum greedy SNR and minimum interference element heuristic scheme for D2D user subchannel allocation. Compared with the minimum interference scheme, the maximum greedy SNR scheme achieved a higher sum rate and lower computational complexity, but required global channel information; the minimum interference element heuristic scheme required the location information of each node. Reference [26] proposed a three-dimensional D2D communication resource allocation scheme based on a three-step hypergraph, which solved the non-convex problem of mixed integers in traditional D2D communication resource allocation, improved the confidentiality rate of the D2D communication system and obtained an approximate optimal result of o (n4) time complexity. Reference [27] proposed a weighted minimization clustering model considering both social attributes and physical proximity, and then used the stable matching theory to optimize the system throughput and achieve one-to-one matching of wireless spectrum resources. Reference [28] proposed an improved interference map resource allocation scheme. Compared with traditional random multiplexing allocation, the centralized allocation of communication resources for detecting packets by the base station (BS) can obtain more useful interference map information and improve the resource allocation capability of cellular networks. Reference [29] proposed a D2D communication mode selection and resource allocation algorithm based on graph theory, and applied it to the full duplex cellular D2D communication system, which improved the utilization of the radio spectrum and maximized the throughput of the communication system. Reference [30] proposed a resource allocation algorithm based on the alternative offer bargaining game. In the whole game, the player is a D2D user, and the reward and punishment return is the transmission power. The adjacent D2D users compete with each other for a higher signal to interference noise ratio (SINR). The algorithm greatly improved the system throughput on the premise of ensuring the user's QoS. Reference [31] proposed a dynamic resource allocation scheme between cells based on repeated games. A pair of D2D users played repeated games with nearby BTSs to maximize the utility function of D2D users and significantly improve the throughput of the entire D2D communication system. Reference [32] proposed a D2D communication resource allocation algorithm based on a Stackelberg game, established a master–slave game model for millimeter wave base stations and D2D users, reduced system energy consumption and improved users' SINR and spectral efficiency. However, in future wireless networks with dense users and rapidly changing scenarios, resource allocation will mainly face two challenges. First, as the number of users increases, acquiring channel state information [33] (CSI) requires a huge signal overhead. It is unrealistic to assume that BS will have global network information. Second, the resource allocation problem is usually modeled as a combination optimization problem with nonlinear constraints, which is difficult to effectively optimize with traditional optimization methods.

Fortunately, deep reinforcement learning (DRL) has been proved to be effective in solving decision-making problems under uncertainty [34]. Reference [35] proposed a DQN-based resource allocation and power control algorithm to maximize system capacity and spectral efficiency, while ensuring sufficient QoS for D2D users. Reference [36] proposed a multi-agent deep reinforcement learning method based on a Stackelberg game, which uses the Stackelberg Q value (ST-Q) to guide the learning direction and performs well in improving the average utility and channel capacity. Reference [37] proposed a resource allocation scheme of multi-agent DQN, which combines DQN with DDPG to reduce the complexity of the network and improve the throughput and fairness of the network. Reference [38] proposed a D2D spectrum access algorithm based on dual-depth Q network (DDQN) to improve the QoS of cellular and D2D users. Reference [39] proposed a D2D communication resource allocation scheme based on dual DQN that can reduce system interference and improve system throughput while obtaining a small amount of channel information. Reference [40] proposed an improved DRL resource allocation scheme to optimize the transmission power of D2D users and cellular users. Reference [41] proposed a 6G-oriented D2D resource allocation scheme based on joint reinforcement learning to ensure the quality of service (QoS) requirements of cellular users and D2D users, while maximizing

the total capacity and minimizing the total power consumption. Reference [42] combined Q learning with an adaptive greedy algorithm to optimize the energy efficiency of D2D in heterogeneous networks, but ignored the convergence of the algorithm. Reference [43] proposed a collaborative reinforcement learning D2D resource allocation scheme, which realized the cooperation of adjacent D2D communication users using shared value function and sharing strategy ideas, and optimized the quality of service and throughput of users in the communication system. Reference [44] proposed a deep reinforcement learning D2D communication resource allocation algorithm based on priority sampling, which helps the data characteristics obtained by the system after agent learning environment interaction to the greatest extent, improves the D2D communication network's resource allocation ability, and reduces network latency. Reference [45] modeled the resource allocation problem in a D2D communication network as a Markov decision problem, realized the adaptive switching between the traditional cellular communication mode and the D2D communication mode, established the actor network and critic network architecture, and improved the energy efficiency of the system. Reference [46] proposed a logarithmic cooling D2D resource allocation scheme based on Q learning to improve the access rate of D2D communication in multiple cells, but ignored the communication quality of users.

However, the above work does not effectively solve the problem of user QoS and throughput in DRL-based D2D communication resource allocation. Especially in the case of extremely low delay requirements, this is still worth further research to improve the performance of D2D and cellular communications.

## 2. System Model

### 2.1. System Model Establishment

In this paper, we consider the multi-D2D multiplexing single cell network communication system in the single cell scenario. Figure 1 shows the system model.

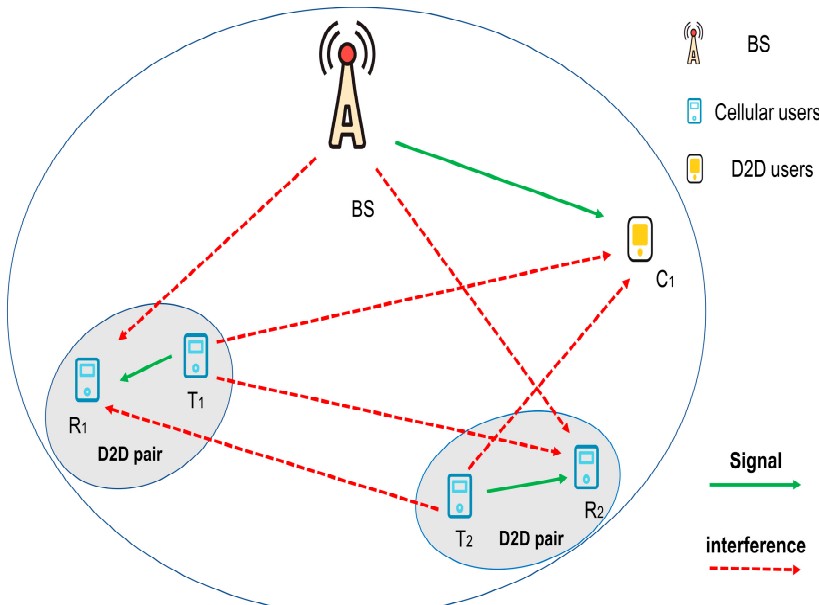

**Figure 1.** D2D communication system model.

The m-th cellular user in the system is defined as $Cm$, where $1 \leq m \leq M$, and the n-th D2D communication pair is defined as $Dn$, where $1 \leq n \leq N$. The D2D communication pair is n, and the transmitting user and receiving user are represented by Tx and Rx, respectively. Both the downlink communication of the cellular link and the D2D link communication use orthogonal frequency division multiplexing (OFDM) technology. Each cellular user occupies a physical resource block RB, and there is no interference between any two cellular links. In the system model, a cellular user is allowed to share the same RB with multiple

D2D users, and the D2D users can independently allocate RB. There are three types of interference in the system model, namely: (1) the cellular users are interfered by D2D transmitting users sharing the same RB; (2) interference from the base station received by the D2D receiving user; (3) interference between D2D links sharing the same RB.

Set the transmission power of the base station to be fixed, expressed by $P_B$. The transmission power of the D2D transmitting user is adjustable, expressed by $P_D$. The channel gain of the cellular downlink target link from the base station to the cellular user $C_m$ and the channel gain of the D2D target link from the D2D transmitting user $D_n^t$ to the receiving user $D_n^r$ are respectively expressed by $G_{B,Cm}$ and $G_{D_n^t,D_n^r}$. When multiple links share RB, the channel gains of interference links from D2D transmitting users to cellular users, from base stations to D2D receiving users, and from D2D transmitting users to receiving users are $G_{D_n^t,C_m}$, $G_{B,D_n^r}$ and $G_{D_i^t,D_i^r}$, respectively.

The signal to interference noise ratio (SINR) ($SINR$) of the received signal on the k-th RB received by the cellular user (Cm) ($C_m$)from the base station can be expressed as:

$$SINR_{Cm} = \frac{P_B G_{B,Cm}}{\sum\limits_{n \in D_k} P_{D_n^t} G_{D_n^t,C_m} + N_0} \tag{1}$$

The SINR of D2D communication to the received signal on the k-th RB can be expressed as:

$$SINR_{D_n} = \frac{P_{D_n^t} G_{D_n^t,D_n^r}}{P_B G_{B,D_n^r} + \sum\limits_{n \in D_k, i \neq n} P_{D_i^t} G_{D_i^t,D_n^r} + N_0} \tag{2}$$

Here $D_k$ represents the set of D2D communication pairs using the k-th RB, and represents the power spectral density of additive white Gaussian noise (AWGN).

Combining the above Formulas (1) and (2) and the Shannon formula, we can obtain the unit bandwidth communication rates of the cellular link and the D2D link respectively:

$$R_{Cm} = \log_2(1 + SINR_{Cm}) \tag{3}$$

$$R_{Dn} = \log_2(1 + SINR_{D_n}) \tag{4}$$

Because cellular users are the primary users of the cellular frequency band, the communication quality of cellular users needs to be guaranteed. The outage probability of the cellular communication link must meet the following conditions:

$$P\left(SINR_{Cm} \leq SINR_C^{tgt}\right) \leq \varepsilon_C \tag{5}$$

This represents the minimum threshold of SINR received by the cellular communication link and the maximum threshold of outage probability of the cellular communication link.

### 2.2. Problem Establishment

After the system model in Figure 1 is established, it is assumed that each cellular user is assigned an RB and the RB will not be shared between cellular users. One RB can be assigned to multiple D2D communication pairs. The RB allocation matrix is defined as $B_{N \times K} = [b_{n,k}]$ to represent the RB allocation of D2D communication pairs. When the k-th RB is assigned $D_n$ to the D2D communication pair, $b_{n,k} = 1$; otherwise, $b_{n,k} = 0$. Define a power control vector $P_N = [P_{D_n^t}]$, where $P_{D_n^t}$ represents the transmission power of the D2D transmitting user $D_n^t$.

The optimization objective of this chapter is to maximize the system capacity by optimizing the RB allocation matrix $B_{N \times K}$ and power control vector $P_N$ of the D2D commu-

nication pair on the premise of ensuring the communication quality of cellular users and D2D users. The optimization problem can be described as follows:

$$\max_{B_{N \times K}, P_N} \left( \sum_{m=1}^{M} R_{Cm} + \sum_{n=1}^{N} \sum_{k=1}^{K} b_{n,k} R_{D_n} \right) \tag{6}$$

$$s.t. \quad P\left( SINR_{Cm} \leq SINR_C^{tgt} \right) \leq \varepsilon_C \tag{7}$$

$$P\left( SINR_{D_n} \leq SINR_D^{tgt} \right) \leq \varepsilon_D \tag{8}$$

$$\sum_{k=1}^{K} b_{n,k} \leq 1, b_{n,k} \in \{0,1\} \tag{9}$$

$$P_{D_n^t} \leq P_{\max} \tag{10}$$

The first constraint Equation (7) makes the outage probability of the cellular link less than a threshold, which is used to ensure the communication quality of cellular users. $SINR_C^{tgt}$ and $\varepsilon_C$ represent the minimum threshold of the cellular communication link reception and the maximum threshold of the cellular communication link outage probability, respectively. The second constraint Equation (8) makes the outage probability of the D2D link less than a threshold to ensure the communication quality of D2D users. $SINR_D^{tgt}$ and $\varepsilon_D$ represent the minimum threshold for the D2D communication link to receive SINR and the maximum threshold for the outage probability of the D2D communication link, respectively. The third constraint Equation (9) indicates that each D2D communication pair can only be assigned one RB at most. The fourth constraint Equation (10) indicates that the transmission power of the D2D transmitting user cannot exceed a maximum transmission power threshold $P_{\max}$.

## 3. Proposed Algorithm

This paper proposes a model of the multi-agent environment, and then proposes a distributed framework based on multi-agent deep reinforcement learning to solve the resource allocation problem of D2D communication.

### 3.1. A2C Environment Model based on Deep Learning

The goal of this paper is to find a resource allocation strategy to maximize the D2D system throughput in cellular systems. In this paper, each D2D is regarded as an agent, and Figure 2 is a model of the A2C-based multi-agent D2D communication system.

Each D2D communication pair acts as an agent, which includes an actor network and a critic network. In a time slot t, the actor network and critic network observe a state St from the state space of the D2D communication environment, and calculate the corresponding mathematical expectation value according to their respective action value function and state value function. Finally, the advantage value calculated by the critic network is used for dynamic action selection of D2D communication. Actions $\alpha_t$ include RB and transmission power selected by the D2D communication pair.

In time slot t, after the agent executes the action, the environment may change, the agent's observation of the environment will shift to a new state $S_{t+1}$ and a reward $r_i$ will be obtained. This reward is determined by the capacity of the D2D communication link corresponding to the agent and the communication quality of the cellular users sharing the spectrum with the D2D communication link. The agent adjusts the new strategy according to the return obtained so that it can obtain a higher return.

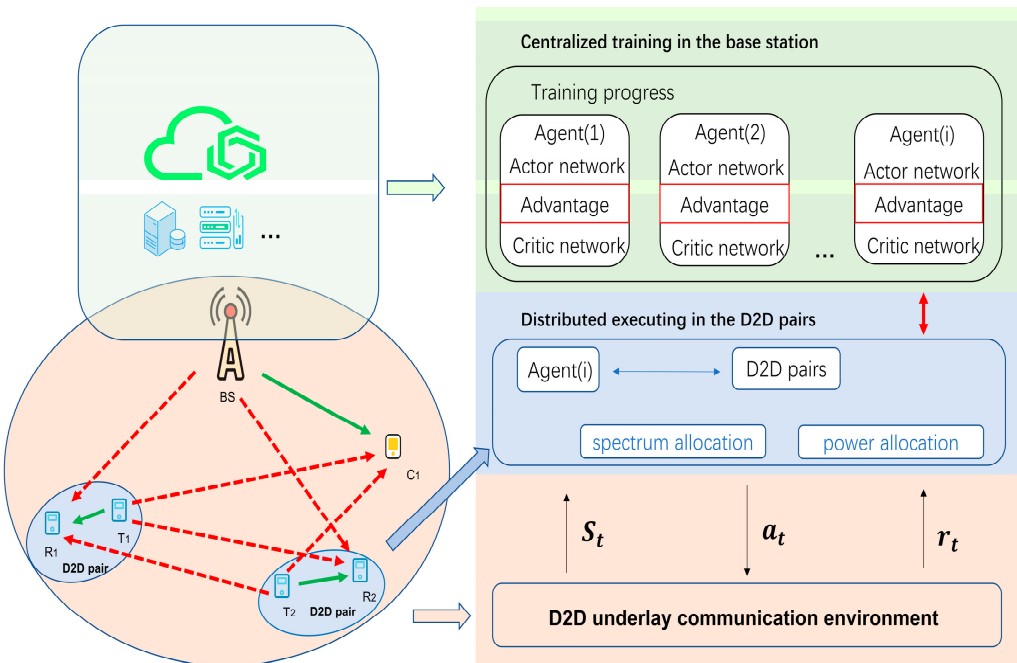

**Figure 2.** Model of A2C-based multi-agent D2D communication system.

Combined with the model, the relevant settings of status, action and reward are as follows:

State space: $S_t^i = \left[ \gamma_i^{D,t}, \gamma_i^{C,t}, I_i^{t-1}, G_t^D, G_t^C, K_{t-1}^C, K_{t-1}^D \right]$

$\gamma_i^{D,t}$ represents the signal-to-noise ratio of the i-th agent at that time; $\gamma_i^{C,t}$ indicates the cellular user's signal-to-noise ratio multiplexed by the i-th agent at the time; $I_i^{t-1}$ indicates the interference to the D2D communication link in the last timeslot; $G_t^D$ indicates the instantaneous channel status information of the D2D communication link; $G_t^C$ indicates the instantaneous channel status information from the BS to the D2D receiving user; $K_{t-1}^C$ indicates the RB occupied by the adjacent cellular users of the D2D communication pair in the last time slot. $K_{t-1}^D$ indicates the RB occupied by the adjacent D2D communication pair of the D2D communication pair in the last time slot.

Action space: $a_t^i = \left[ P_i^t, B_i^t \right]$. $P_i^t$ represents the power selection of the i-th agent at time t; $B_i^t$ represents the channel resources multiplexed by the i-th agent at time t.

Reward function: $r_t^i = \left[ R_D^t, R_C^t \right]$. $R_D^t$ represents the throughput of D2D users of the i-th agent at time t; $R_C^t$ indicates the cellular user throughput at time t.

The multi-agent D2D deep reinforcement learning communication system model is the extension of the single-agent model. This paper proposes a multi-agent communication system with collective training and decentralized execution, as shown in Figure 2. The multi-agent model includes multiple single agents, and each single agent is composed of actor network, critic network and advantage function. The actor network selects actions according to the state information observed by the agent, and the critic network gives a preliminary score of the original operation's advantages and disadvantages after performing the actions. Finally, the advantage function is used to dynamically adjust the selection actions. Both actor network and xritic network are fitted with a depth neural network, as shown in Figure 3.

### 3.2. Network Parameter Update

The A2C algorithm is mainly composed of three parts, namely, dominance function, actor network and critic network. The dominance function is shown in Figure 3. If the action exceeds the average performance, the output value of the dominance function is positive,

and vice versa. In the multi-agent actor–critic deep reinforcement learning network, the advantage function of the i-th D2D user is:

$$A^i\left(s_t^i, a_t^i\right) = Q^i\left(s_t^i, a_t^i\right) - V^i\left(s_t^i\right) = E\left[r_t^i \middle| s_t^i, a_t^i\right] - V^i\left(s_t^i\right)$$
$$\approx r_t^i + \gamma V^i\left(s_{t+1}^i \middle| s_t^i, a_t^i\right) - V^i\left(s_t^i\right) = \delta^i\left(\delta_t^i\right) \tag{11}$$

In the formula $A^i = \left(s_t^i, a_t^i\right)$, $Q^i = \left(s_t^i, a_t^i\right)$, $V^i = \left(s_t^i\right)$, $\delta^i = \left(\delta_t^i\right)$ and $V^i = \left(s_{t+1}^i \middle| s_t^i, a_t^i\right)$ are respectively expressed as the time advantage function, action value function, state value function, TD error value and state value function at t+1 of the i-th agent.

For parameter update in the actor network, we adopt TD error based on gradient rise to update.

$$\theta_{t+1}^i = \theta_t^i + \alpha_a \nabla_{\theta^i} J\left(\theta_t^i\right) \tag{12}$$

$\alpha_\alpha$ indicates the learning rate of the agent. $\nabla_{\theta^i} J\left(\theta^i\right) \approx \delta^i\left(s_t^i\right) \nabla_{\theta^i} \ln \pi_{\theta^i}^i\left(a_t^i \middle| s_t^i\right)$, $\theta^i$ indicates the policy network parameters of the i-th agent. $\pi_{\theta^i}^i\left(a_t^i \middle| s_t^i; \theta_i\right)$ represents the policy of the i-th agent.

The loss function of the critic network of the i-th agent is defined as:

$$L_{Critic}^i = \left[\delta^i\left(s_t^i\right)\right]^2 = \left(r_t^i + \gamma V^i\left(s_{t+1}^i \middle| s_t^i, a_t^i\right) - V^i\left(s_t^i\right)\right)^2 \tag{13}$$

Similarly, the critic network uses a gradient rise algorithm to update its own parameters:

$$w_{t+1}^i = w_t^i + \alpha_c \nabla_{w^i} V_{w^i}^i\left(s_t^i\right) \delta^i\left(s_t^i\right) \tag{14}$$

Both actor network and critic network adopt DNN parameter optimization. The actor network output layer uses the Softmax activation function to determine the probability of each action. The critic network is used to assist strategy training, help the evaluation of the superiority function and output the current optimal strategy. The flow of the algorithm proposed in this paper is shown in Algorithm 1.

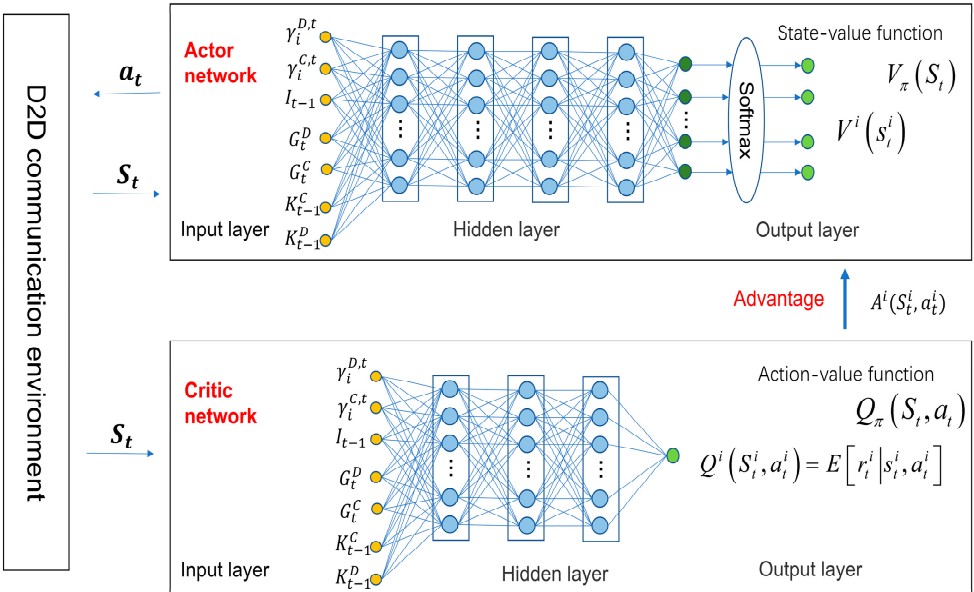

**Figure 3.** A2C communication model based on deep learning.

| **Algorithm 1.** D2D communication resource allocation algorithm |
| --- |
| Algorithm: A D2D Communication Resource Allocation Algorithm Based on A2C |
| Initialization: |
| 　　　　　　　　Initialize the cell, base station, cellular user and D2D user using communication simulation.<br>$\pi$　　　　　　　　: Policy model for all D2D users $\pi$<br>Actor network: Parameter $\theta$, Learning rate $\alpha_a$<br>Critic network: Parameter $\omega$, Learning rate $\alpha_c$<br>$\gamma$　　　　　　　　: Discount factor $\gamma$<br>Advantage function: $A_t^i\left(S_t^i, a_t^i\right)$<br>T　　　　　　　　: Number of communication simulation timeslot cycles. |
| Train : |
| 　　　　1: $t \leftarrow 0$<br>　　　　2: Cycle:<br>　　　　3: All D2D communication users observe their own state $S_t^i$.<br>　　　　4: All D2D communication users are based on the current state $S_t^i$ and Policy $\pi$. Output $a_t^i$, $R_B$ and transmit power $P_t^i$.<br>　　　　5: All D2D communication users are based on the current status $S_t^i$. and $\pi$. $R_t^i$ Rewards obtained by output $a_t^i$.<br>　　　　6: All D2D communication users observe the next state $S_{t+1}^i$<br>　　　　7: All D2D communication users input $S_{t+1}^i$ into the critic network as an input parameter and obtain the mathematical expectation of the critic network to calculate the advantage function.<br>　　　　8: Update Critic network parameters $\omega$.<br>　　　　9: Update actor network parameters $\theta$.<br>　　　　10: Update MAA2C algorithm strategy $\pi$.<br>　　　　11: $R_{t+1}^i$ Rewards obtained by output $a_{t+1}^i$.<br>　　　　12: $t \leftarrow t + 1$ Simulation platform updates.<br>　　　　13: Until t = T, return the test result. |

## 4. Simulation Results and Analysis

In order to enrich the reliability and progressiveness of the experiment, this paper takes the outage probability and throughput of users in the D2D communication network as the experimental indicators. The interruption probability is an expression of the link capacity. When the link capacity cannot meet the required user rate, communication interruption will occur, and the event probability is the interruption probability. Throughput can reflect the ability of the network to transmit data. In the algorithm comparison phase, this paper uses two dimensions for comparison. First, DQN, which belongs to the field of deep reinforcement learning, is compared with the multi-agent D2D communication resource allocation algorithm (MAA2C) proposed in this paper based on A2C, to further prove the reliability of deep reinforcement learning in dealing with D2D communication resource allocation problems. Then, MAAC, which is also based on actor–critic architecture, is used as the comparison algorithm to horizontally compare progressiveness. The data indicators of the comparison experiment include the convergence of the algorithm, the user's outage probability and the system throughput. Experiment simulation parameters and simulation experiment diagram are shown in Algorithm 1and Figure 4, respectively.

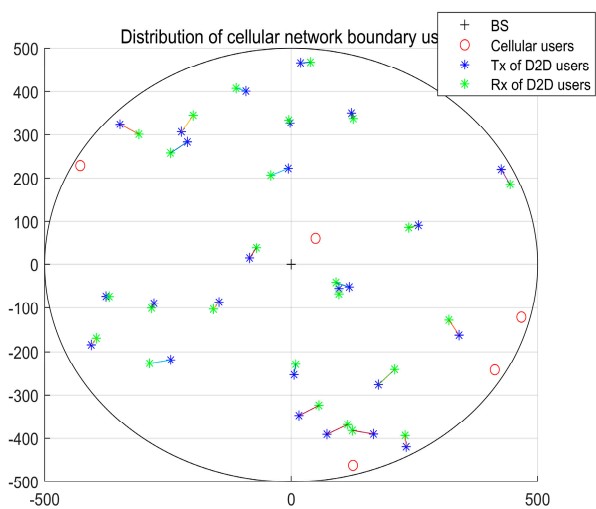

**Figure 4.** Distribution of cellular network boundary users.

*4.1. Simulation Parameter Setting*

For the setting of simulation experiment parameters, this paper considers a cell with a radius of 500 m. The cellular users and D2D communication pairs are randomly distributed in the cell. The number of D2D users is 5, 10, 15, 20 or 25. The number of cellular users is 5. The threshold of outage probability for each D2D cell user is 0.01. The noise power spectral density is −174 dBm/Hz. See Table 1 for specific simulation parameters.

**Table 1.** Simulation Parameters.

| Simulation Parameters | Parameter Value (unit) |
|---|---|
| Base station transmit power | 46 dBm |
| D2D transmits the maximum transmit power of the user | 23 dBm |
| D2D communication to the maximum distance | 20 m |
| Number of cellular users | 5 |
| The number of RB | 5 |
| Number of D2D communication pairs | 5, 10, 15, 20, 25 |
| Path loss model of cellular communication link | $128.1 + 37.6log_{10}(d)$ |
| D2D communication link path loss factor | 4 |
| Cellular user target SINR threshold | 0 dB |
| Cellular subscriber outage probability threshold | 0.01 |
| D2D user target SINR threshold | 0 dB |
| D2D user outage probability threshold | 0.01 |
| Noise power spectral density | −174d dBm/Hz |

The cellular network and the distribution of users are shown in Figure 4. The simulation experiment is a unit circle with a radius of 500, in which the red circle is the cellular user, the middle is the base station position and the blue and green divergent circles are the positions of the transmitter and receiver of the D2D user.

| parameter | value |
|---|---|
| Base station transmit power | 46dBm |
| D2D transmits the maximum transmit power of the user | 23dBm |
| D2D communication to the maximum distance | 20m |
| Number of cellular users | 5 |
| The number of RB | 5 |
| Number of D2D communication pairs | 5,10,15,20,25 |
| Path loss model of cellular communication link | $128.1 + 37.6\log_{10}(d)$ |
| D2D communication link path loss factor | 4 |
| Cellular user target SINR threshold | 0dB |
| Cellular subscriber outage probability threshold | 0.01 |
| D2D User target SINR threshold | 0dB |
| D2D user outage probability threshold | 0.01 |
| Noise power spectral density | -174dBm/Hz |

### 4.2. Result Analysis

Deep reinforcement learning is one of the important ways to deal with the D2D communication resource allocation scheme, and has attracted much attention. How can one judge the quality of the deep reinforcement learning algorithm? The evaluation index of convergence gives a perfect answer. In this part, three reinforcement learning algorithms, MAA2C, MAAC and DQN, are simulated and compared in the same experimental environment. Figure 5 shows the trend of all agents' total benefit parameters and iterative convergence in the single cell and five D2D users' multiplexing mode. It can be seen from the figure that the DQN algorithm has the fastest convergence speed, but the overall throughput benefit is not good. This is because compared with other algorithms, the DQN algorithm unilaterally overemphasizes the competition between agents and the environment and the interaction of environmental information, ignoring the idea of cooperation. The MAAC algorithm has the worst convergence. This is because, compared with MAA2C, the MAAC algorithm is slow in adjusting cooperation ideas each time. There is no system advantage function that can directly fine tune the actor network and critic network. Looking at the three deep reinforcement learning algorithms, it is not difficult to find that MAA2C has the best effect, whether it is the number of convergence iterations or the income of the algorithm after the stable iteration.

In order to prove that the proposed algorithm has good performance in protecting the communication quality of cellular users, this paper presents the curve trend chart of cellular user outage probability. Figure 6 shows the relationship between the outage probability of cellular users and the number of D2D links. It can be seen from the figure that with the increase in D2D logarithm, the distributed MAA2C algorithm proposed in this paper is superior to the other two algorithms. First, the three methods all use the distributed interference management mode, so the curve trend is similar. Secondly, the MAAC algorithm is better than the DQN algorithm because there is a cooperation mechanism in MAAC. The performance of the MAA2C algorithm is better than that of the MAAC algorithm, because MAA2C not only has a cooperation mechanism, but also has an advantage function mechanism, which enables multiple D2D users to consider cooperation in the reuse competition scenario to generate higher benefits. On the other hand, MAA2C uses the adjustment of the advertising advantage function to perform more accurate proxy actions.

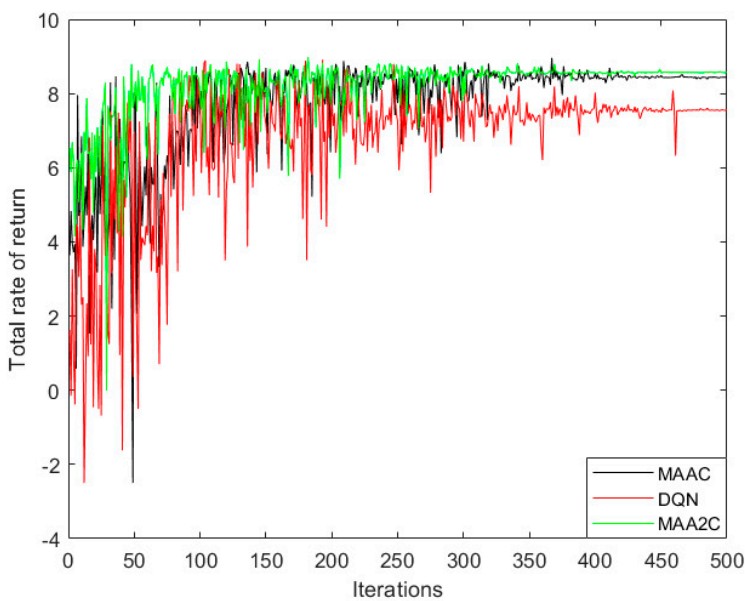

**Figure 5.** Trend chart of convergence of total return.

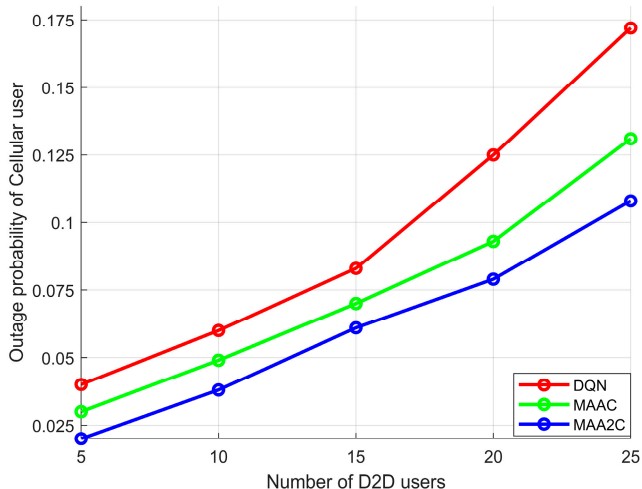

**Figure 6.** Outage probability of cellular users.

In order to prove that the D2D link has good communication performance when the algorithm proposed in this paper is multiplexed in cellular networks, the trend curve of D2D outage probability is shown. Figure 7 shows the relationship curve between the outage probability of D2D users and the number of D2D users. It can be seen from Figure 7 that the algorithm proposed in this paper is better than the DQN resource allocation scheme, and slightly better than MAAC. It can be seen from the figure that the outage probability of DQN is the highest. The performance of MAAC algorithm is better than that of DQN, because the MAAC algorithm introduces global information to guide training, which enables agents to learn cooperative strategies and avoid interference caused by multiple D2D users' competitive access during distributed execution. The performance of the MAA2C algorithm is better than that of MAAC, because MAA2C fully considers the parameter updating process of the state value network and action value network in the AC network and the influence of the difference between them, i.e., the dominance function, on the system on the basis of considering the cooperation of multiple D2Ds.

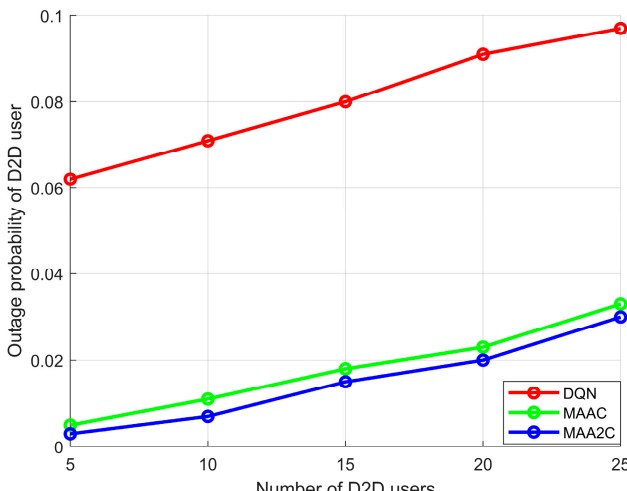

**Figure 7.** Outage probability of D2D users.

In order to prove that the proposed algorithm has good performance for D2D user throughput in cellular networks, Figure 8 shows the change curve of the number of D2D link systems and D2D communication pairs. The total throughput of the D2D link system is represented by the ordinate (after the throughput of all D2D users). The abscissa represents the change in the number of D2D users. It can be seen from the figure that the MAA2C algorithm performs best among the three algorithms. It can be seen from the figure that when the number of D2D users is two to three times the number of cellular users, the optimization effect gradient is MAA2 > MAAC > DQN. This is because the reward function in the MAA2C algorithm is based on the real-time throughput of D2D users in the D2D link, and the throughput growth of D2D users is taken as the positive feedback of the system. The effect of all outputs is optimal. In the later period, as the number of D2Ds continues to increase, the algorithm proposed in this paper grows slowly. This also proves that the proposed algorithm has good convergence ability and dynamic adaptive resource allocation ability.

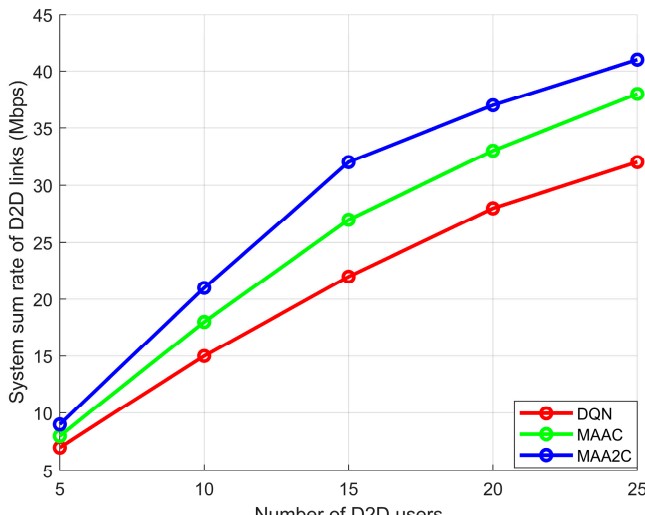

**Figure 8.** System sum rate of D2D links (Mbps).

The throughput output capability of a system is an important indicator to measure the data communication capability of a communication system. In order to prove that the proposed algorithm has good D2D communication resource allocation capability, this paper simulates the trend curve of the throughput output of the cellular system. The system

capacity includes the communication capacity of D2D users and cellular users. Figure 9 shows the curve of system capacity changing with the number of D2D communication pairs. It can be seen from the figure that compared with the DQN algorithm and the MAAC algorithm, the MAA2C algorithm in this paper has obvious advantages. The first point is the system throughput. It can be seen that the performance is good, because the MAA2C algorithm can learn and cooperate during centralized training, as well as rationalize the dynamic selection of power and channel, so as to obtain the maximum global total return. Compared with other distributed algorithms, the algorithm proposed in this paper can make decisions that are more conducive to improving the global performance. Second, in terms of the system convergence performance, the comparison of the three algorithms shows that the MAA2C algorithm has a smooth output curve, while the other two have large fluctuations, which fully demonstrates that the MAA2C algorithm has good convergence. In conclusion, the algorithm proposed in this paper is optimal both in terms of algorithm convergence ability and system throughput output in a D2D system.

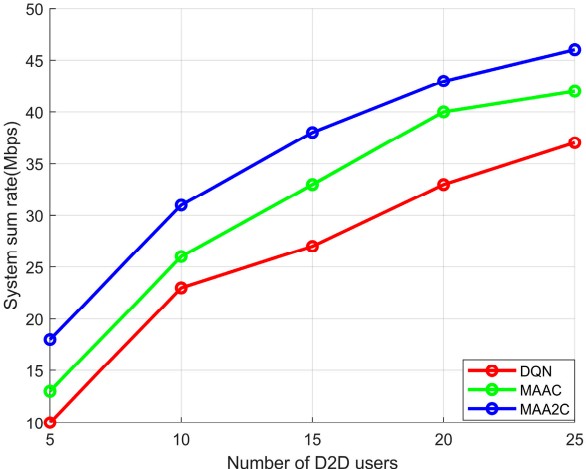

**Figure 9.** System sum rate (Mbps).

## 5. Conclusions

In this paper, a multi-agent D2D communication resource allocation algorithm based on A2C is proposed. The D2D user is regarded as an agent. Through the interaction information between the multi-agent and the wireless communication environment, the policy network and the state network are deeply studied and trained. Finally, the best transmission power output and the best channel matching of the D2D user are achieved. Simulation results show that the proposed algorithm has the best performance in improving system throughput and reducing user interruption. At the same time, we found that compared with a single policy network (DQN), the actor–critic dual network has a higher system performance optimization capability in D2D resource allocation. Compared with MAAC and MAA2C, we can see that a certain degree of actor–critic complex network has a better D2D communication resource allocation capability. However, deep reinforcement learning itself has the problems of slow training speed and difficult convergence, which requires further optimization of learning efficiency. In the future, we will continue to deepen the complexity of the actor–critic network and further improve the convergence ability of the algorithm.

## 6. Patents

A patent entitled "Heterogeneous cognitive wireless sensor network cluster routing method" is disclosed under CN110708735B.

**Author Contributions:** Conceptualization, X.L. and G.C. (Guifen Chen); methodology, X.L.; validation, X.L., G.C. (Guifen Chen) and G.C. (Guangjiao Chen); data curation, X.L.; writing—original draft preparation, X.L.; writing—review and editing, X.L.; visualization, X.L.; project administration, G.C. (Guifen Chen), G.W. and Z.S.; funding acquisition, G.C. (Guifen Chen). All authors have read and agreed to the published version of the manuscript.

**Funding:** This research was funded by "Thirteenth Five-Year Plan" Science and Technology Research Project of Jilin Provincial Department of Education, Research on Large-scale D2D Access and Traffic Balancing Technology for Heterogeneous Wireless Networks JJKH20181130KJ, Special Project on Industrial Technology Research and Development of Jilin Province, Research on Self-organizing Network System of Unmanned Platform for Optoelectronic Composite Communication, 2022C047-8.

**Acknowledgments:** The authors acknowledge the National Natural Science Foundation of China [grant number 61540022] and the Key R & D projects of Changchun Science and Technology Bureau [grant number 21ZGM43].

**Conflicts of Interest:** The authors declare no conflict of interest.

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
