# Peer review of "Research on Multi-Agent D2D Communication Resource Allocation Algorithm Based on A2C"

_electronics, doi:10.3390/electronics12020360_

Round 1
Reviewer 1 Report
The authors presented a study under title of “Research on Multi-Agent D2D communication resource allocation algorithm based on A2C". The applied methodology is sound. Also, the study looks good. Nonetheless, some issues need to be addressed.
Issues, weaknesses:
1. What kinds of training data sets are used to train the model through reinforcement learning process?
2. How could the authors tackle the catastrophic forgetting and handling imbalance data problems during the training?
3. The authors missed to explain the practical compatibility of the D2D communication with Machine learning approaches. The authors should fix it.
4. What necessary actions should be taken by the authors to make this D2D communication as a supervised approach?
5. Mention of the motivation of the proposed model and highlight it in the abstract.
6. What is the role of A2C, DQN and MAAC? Clarity Needed.
7. More literature survey is required.
8. More discussion on results is needed.
9. There is much abbreviation in abstract like A2C, DQN and MAAC. Please a full name.
10. Please uniform the size of every equation.
11. Please make Fig. 3. (A2C Communication Model Based on Deep Learning) and Fig. 5. (Trend Chart of Convergence of Total Return) more clear.
12. Extensive English language corrections are needed.
13. Need thorough revision and proper structuring of the research article.
Author Response
Dear Reviewers:
Thank you very much for consulting my manuscript in your busy schedule and giving me many opinions that have benefited me greatly. These suggestions have greatly improved my ability to write manuscripts and my understanding of D2D communication, which is of great help to my scientific research. I have carefully studied the excellent opinions you have given, and made profound corrections on the premise of ensuring that the logic of the full text framework remains unchanged.
Thank you again for your letter and the reviewer's comments on our manuscript entitled " Research on Multi-Agent D2D communication resource allocation algorithm based on A2C " (ID: electronics-2127221). These comments are valuable, help to modify and improve our paper, and have important guiding significance for our research. We have carefully studied the opinions and made corrections, hoping to get approval. The modified part is marked in red on the paper. The main corrections and responses to reviewer comments in the paper are as follows:
Reviewer #1:
- Response to comment1:
Response:
No specific data set is used in the manuscript, because the data used in this paper is generated by the agent's continuous interaction with the environment. The training data set adopted in this paper consists of the agent's signal-to-noise ratio, the agent's signal-to-noise ratio when reusing cellular users, the interference state of the D2D communication link, the instantaneous channel state of the D2D communication link, the channel state from the base station to the D2D receiver, the state of the resource block occupied by the cellular users adjacent to the D2D users, and the state of the resource block occupied by the D2D users adjacent to the D2D users; The tag data is a time-based delay reward, that is, the throughput of the optimal user in the system and the optimal D2D resource allocation scheme.
- Response to comment2:
Response:
Aiming at the catastrophic forgetting problem of multi-agent (multi task), this paper allocates a memory to each agent, stores the scenario samples of the corresponding agent interacting with the environment in each memory, defines the loss function, and then solves the optimal network gradient of the model based on the network gradient of the current task and the previous task. Finally, the neural network in the fusion network gradient solution and reinforcement learning method is used, Redefine the loss function of the network, update the Q value of the reinforcement learning training process to realize the network gradient in the network and Q value estimation network, so as to alleviate catastrophic forgetting.
For the processing of unbalanced data, this paper uses the penalty weight of positive and negative samples to solve the problem of sample imbalance. Different weights are assigned to categories with different sample numbers in training (small sample influence factor weights are slightly higher, large sample influence factor weights are slightly lower).
- Response to comment3:
Response:
With regard to compatibility, this paper introduces the application of reinforcement learning in D2D communication at home and abroad. Relevant literature has proved that machine learning algorithm is feasible in D2D communication. “Fortunately, deep reinforcement learning (DRL) has been proved to be effective in solving decision-making problems under uncertainty [28]. Reference [29] proposed a DQN based resource allocation and power control algorithm to maximize system capacity and spectral efficiency, while ensuring sufficient Qos for D2D users.”
- Response to comment4:
Response:
In this paper, D2D communication is approximately regarded as a supervision method by setting parameters in deep reinforcement learning. The specific parameter settings are as follows:
(1) tolerance parameters:tolerance grad=1e-05, tolerance change=1e-09,
(2) hyper parameter:
In the system throughput calculation, the learning rates of Actor and Critical networks are set to 0.0001 and 0.001 respectively; Discount factor =0.95. The Actor network is composed of three hidden layers. Each hidden layer has 64 neurons and an ELU activation function. The number of neurons in the output layer is the number of all optional actions of the agent. Then, the output results are passed through the Soft max activation function to obtain the probability distribution of actions. Critical network includes two hidden layers, each hidden layer has 64 neurons and an ELU activation function, and the output layer has only one neuron to provide V (s) estimation of the state value function.
In the calculation of outage probability, the learning rates of Actor and Critical networks are set to 0.0001 and 0.0001 respectively; Discount factor =0.96. The Actor network is composed of two hidden layers. Each hidden layer has 64 neurons and an ELU activation function. The number of neurons in the output layer is the number of all optional actions of the agent. Then, the output results are passed through the Soft max activation function to obtain the probability distribution of actions. Critical network includes three hidden layers, each hidden layer has 64 neurons and an ELU activation function, and the output layer has only one neuron to provide V (s) estimation of the state value function.
- Response to question5:
Response:
The abstract has been revised and marked in red in the new manuscript.
- Response to comment6:
Response:
With regard to the role of A2C, DQN and MAAC algorithms, we made changes in the latest manuscript simulation experiment, and used red font to mark new manuscripts.
- Response to comment7:
Response:
In the introduction, we added the latest literature related to the theme of the article to enrich the content of our article.
- Response to comment8:
Response:
The conclusion has been revised and marked in red in the new manuscript.
- Response to comment9:
Response:
The full names of A2C, DQN and MAAC have been changed in the abstract and marked in red in the new manuscript.
- Response to comment10:
Response:
The equations in the text have been changed and marked in red in the new manuscript.
- Response to comment11:
Response:
Figures 3 and 5 have been changed and marked in red in the new manuscript.
- Response to comment12:
Response:
Thank you for your valuable and thoughtful comments. We have carefully checked and improved the English writing in the revised manuscript.
- Response to comment13:
Response:
Thank you very much for your good comments. We have revised the whole text. The summary and conclusion are revised.
Yours sincerely,
Xinzhou Li
Reviewer 2 Report
The manuscript proposes an A2C-based multi-agent D2D communication resource allocation algorithm. By performing A2C modeling and updating parameters on a multi-D2D-based cellular D2D communication system, the transmission power and resource allocation of D2D users are dynamically adaptively output, which reduces the probability of user interruption and improves system throughput, which is innovative. In addition, the authors conduct extensive experiments and show visualization results to verify the effectiveness of the proposed method. The manuscript has a novel approach and a clear structure, but there are still some issues, therefore, I suggest that the manuscript be accepted after revising the following issues:
(1)The author's discussion of the latest research in the introduction is not sufficient, and it is recommended to add the following articles as references:
1.https://doi.org/10.1109/ACCESS.2020.2982443
(Maize Leaf Disease Identification Based on Feature Enhancement and DMS-Robust Alexnet)
2.https://doi.org/10.1016/j.compag.2020.105730
(Identification of tomato leaf diseases based on combination of ABCK-BWTR and B-ARNet)
3.https://doi.org/10.3389/fpls.2022.846767
(CASM-AMFMNet: A Network Based on Coordinate Attention Shuffle Mechanism and Asymmetric Multi-Scale Fusion Module for Classification of Grape Leaf Diseases)
(2) All the experimental indicators in the manuscript do not explain the meaning, it is recommended to add.
(3) In 4.2 Result analysis, there is no brief introduction to the two algorithms MAAC and DQN compared, and it is recommended to supplement.
(4) In 4.2 Result analysis, only MAAC and DQN were compared in the experiment, and there were few algorithms compared, which could not prove the advanced nature of the proposed method. It is suggested to supplement.
Author Response
Dear Reviewers:
Thank you very much for consulting my manuscript in your busy schedule and giving me many opinions that have benefited me greatly. These suggestions have greatly improved my ability to write manuscripts and my understanding of D2D communication, which is of great help to my scientific research. I have carefully studied the excellent opinions you have given, and made profound corrections on the premise of ensuring that the logic of the full text framework remains unchanged.
Thank you again for your letter and the reviewer's comments on our manuscript entitled " Research on Multi-Agent D2D communication resource allocation algorithm based on A2C " (ID: electronics-2127221). These comments are valuable, help to modify and improve our paper, and have important guiding significance for our research. We have carefully studied the opinions and made corrections, hoping to get approval. The modified part is marked in red on the paper. The main corrections and responses to reviewer comments in the paper are as follows:
Reviewer #2:
- Response to comment1:
Response:
Thank you very much for your comments on the introduction. We have benefited a lot. We have made detailed modifications to the introduction and added the above articles to enrich our manuscript.
- Response to comment2:
Response:
As for the experimental indicators and their meanings, we supplemented them in the latest manuscript simulation experiment, and used red font to mark new manuscripts.
- Response to comment3:
Response:
About the introduction and role of DQN and MAAC algorithms, we made changes in the latest manuscript simulation experiment, and used red font to mark new manuscripts.
- Response to comment4:
Response:
Thank you very much for your valuable comments. We have made changes in the simulation experiment section and explained the original intention of the selected comparison algorithm. We are sorry for the inconvenience.
Yours sincerely,
Xinzhou Li
Reviewer 3 Report
The authors propose a research method to improve the multi-agent device to device communication. Although the results look promising, I have some doubts and hope authors could provide some clarifications.
1. There are some typos and grammar errors in the writing. The manuscript is also not organized well. Some of equations do not have correct mathematical formats. Please spend time to revise the manuscript.
2. Please use the whole name of A2C, DQN and MAAC in the abstract, otherwise people won’t understand these terms.
3. Tables and figures are not clear. Please provide clear version and correct the format issue. And please add more explanation on the legend so people can understand.
4. Please highlight the contributions of this paper.
Author Response
Dear Reviewers:
Thank you very much for consulting my manuscript in your busy schedule and giving me many opinions that have benefited me greatly. These suggestions have greatly improved my ability to write manuscripts and my understanding of D2D communication, which is of great help to my scientific research. I have carefully studied the excellent opinions you have given, and made profound corrections on the premise of ensuring that the logic of the full text framework remains unchanged.
Thank you again for your letter and the reviewer's comments on our manuscript entitled " Research on Multi-Agent D2D communication resource allocation algorithm based on A2C " (ID: electronics-2127221). These comments are valuable, help to modify and improve our paper, and have important guiding significance for our research. We have carefully studied the opinions and made corrections, hoping to get approval. The modified part is marked in red on the paper. The main corrections and responses to reviewer comments in the paper are as follows:
Reviewer #3:
- Response to comment1:
Response:
Thank you very much for your valuable comments. We apologize for the problem of writing English expressions and mathematical formulas. The latest manuscript has completed the changes of English and mathematical formulas.
- Response to comment2:
Response:
The full names of A2C, DQN and MAAC have been changed in the abstract and marked in red in the new manuscript.
- Response to comment3:
Response:
The tables and pictures in the text have been modified. We apologize for the difficulties caused by reading and have updated them in the latest version.
- Response to comment4:
Response:
The conclusion has been revised and marked in red in the new manuscript.
Yours sincerely,
Xinzhou Li
Round 2
Reviewer 1 Report
Accept in Current Condition. No further comment.
Reviewer 2 Report
No more suggestion
Reviewer 3 Report
The revision has improved the quality of the manuscript, but the paper is still not organized well. Please recheck the format (manuscript, table and figures), writing style and grammar errors to meet the requirements.